# Comparison of the Effects of Three Dual-Nucleos(t)ide Reverse Transcriptase Inhibitor Backbones on Placenta Mitochondria Toxicity and Oxidative Stress Using a Mouse Pregnancy Model

**DOI:** 10.3390/pharmaceutics14051063

**Published:** 2022-05-15

**Authors:** Kayode Balogun, Lena Serghides

**Affiliations:** 1Saskatchewan Health Authority, Regina, SK S4S 0A5, Canada; kayode.balogun@saskhealthauthority.ca; 2Department of Pathology and Laboratory Medicine, University of Saskatchewan, Saskatoon, SK S7N 0W8, Canada; 3Toronto General Hospital Research Institute, University Health Network, Toronto, ON M5G 1L7, Canada; 4Department of Immunology and Institute of Medical Sciences, University of Toronto, Toronto, ON M5S 1A1, Canada

**Keywords:** zidovudine, tenofovir, abacavir, emtricitabine, lamivudine, HIV antiretroviral, placenta, oxidative stress, mitochondrial toxicity, pregnancy outcomes

## Abstract

Nucleos(t)ide reverse transcriptase inhibitors (NRTIs) are the backbone of HIV antiretroviral therapy (ART). ART use in pregnancy has been associated with adverse birth outcomes, in part due to NRTI-induced mitochondrial toxicity. Direct comparison on the effects of commonly used dual-NRTI regimens on placental mitochondria toxicity in pregnancy is lacking. We compared zidovudine/lamivudine, abacavir/lamivudine, and tenofovir/emtricitabine using a mouse model and examined markers of placental mitochondrial function and oxidative stress. Zidovudine/lamivudine and abacavir/lamivudine were associated with lower fetal and placental weights compared to controls, whereas tenofovir/emtricitabine was associated with the least fetal and placental weight reduction, as well as lower resorption rates. Placental mitochondrial DNA content, as well as placental expression of cytochrome c-oxidase subunit-II, DNA polymerase gamma, and citrate synthase, was higher in tenofovir/emtricitabine-treated mice compared to other groups. Zidovudine/lamivudine-treated mice had elevated malondialdehyde levels (oxidative stress marker) compared to other groups and lower mRNA levels of manganese superoxide dismutase and peroxisome proliferator-activated receptor gamma coactivator 1-alpha in the placenta compared to tenofovir/emtricitabine-treated mice. We observed differences in effects between NRTI regimens on placental mitochondrial function and birth outcomes. Tenofovir/emtricitabine was associated with larger fetuses, increased mtDNA content, and higher expression of mitochondrial-specific antioxidant enzymes and mitochondrial biogenesis enzymes, whereas zidovudine/lamivudine was associated with markers of placental oxidative stress.

## 1. Introduction

Nucleos(t)ide reverse transcriptase inhibitors (NRTIs) are the mainstay of combination antiretroviral therapy (ART). NRTI-based ART regimens have been successfully used to control HIV infection and prevent perinatal transmission of HIV. All currently recommended ART regimens consist of a dual-NRTI backbone in combination with a protease inhibitor (PI), a non-NRTI (NNRTI), or an integrase strand transfer inhibitor (INSTI) [1,2]. NRTIs are also used in pre-exposure prophylaxis (PrEP) to help prevent HIV infection.

The health benefits of NRTI-based ART are undeniable; however, the prolonged use of NRTIs has been associated with adverse metabolic effects, such as dyslipidemia, lipodystrophy, hepatic steatosis, and lactic acidosis [3]. These side effects have been linked to NRTI-induced mitochondrial toxicity [4,5,6,7,8] and oxidative damage to mitochondrial DNA (mtDNA) [9,10]. 

Mitochondria are vital for cellular activities and functions, such as energy production, metabolism, thermogenesis, and oxidative phosphorylation [11]. Mitochondrial DNA replication and function is regulated by DNA-polymerase gamma (POLG) [12]. A downregulation in the expression of POLG results in impaired production of mtDNA, consequently altering the levels of key proteins involved in oxidative phosphorylation (OXPHOS) [13,14]. 

NRTIs, once intracellularly phosphorylated, compete with the native nucleotide pool to be incorporated into pro-viral DNA during its replication by HIV reverse transcriptase and promote early termination. NRTIs can also serve as substrates for POLG, which can hamper its mtDNA replication and repair activities, resulting in a multifactorial mitochondrial toxicity and altered mitochondrial function [7,14,15,16,17]. The physiological and biochemical consequences of mitochondrial toxicity include alterations in mitochondrial morphology, a reduction in mtDNA, disruption of OXPHOS, and increased reactive oxygen species (ROS) and oxidative stress [14,18]. Mitochondrial biogenesis can also be affected by mitochondrial toxicity via the downregulation of peroxisome proliferator-activated receptor gamma coactivator 1-alpha (PGC-1α), a key regulator of mitochondrial biogenesis [19,20]. 

NRTIs are permeable to the placenta, and there are existing concerns about NRTI-induced mitochondrial toxicity in the placenta during pregnancy [21]. Efficient mitochondrial function is crucial to pregnancy and fetal development [22], and mitochondria toxicity during pregnancy could significantly alter the course of pregnancy and lead to adverse pregnancy outcomes, including fetal growth restriction. Human and animal studies have suggested that exposure to NRTIs during pregnancy can lead to mitochondrial toxicity in the offspring, although clinical findings of mitochondrial toxicity are infrequent [6,23,24,25,26,27,28,29,30,31,32,33]. Symptoms of hyperlactatemia with signs of neurological or cardiac disorders associated with mitochondrial dysfunction, alterations in mitochondrial morphology, and impaired OXPHOS have also been documented in infants exposed to NRTIs in utero [6,28,29,30,32,34]. 

The aim of this study was to compare the effects of commonly prescribed dual-NRTI backbones zidovudine/lamivudine (AZT/3TC), abacavir/lamivudine (ABC/3TC), tenofovir/emtricitabine (TDF/FTC) on placental mitochondrial toxicity during pregnancy, by evaluating the impact of these NRTI combinations on mtDNA content and genes involved in mitochondria-mediated antioxidant defense systems, mitochondrial biogenesis, and mitochondrial function as markers of mitochondrial toxicity. The objective of this study was to directly compare these NRTI backbones in a murine pregnancy model with the goal of determining which combination is associated with the best pregnancy outcomes and the least mitochondrial toxicity. 

## 2. Materials and Methods

### 2.1. Animals and Protocols

All animal studies were performed in accordance with the Canadian Council on Animal Care guidelines and were approved by the University Health Network Animal Use Committee. The housing conditions, breeding, and drug treatment of the mice are described in our previous publication [35]. Briefly, C57BL/6 mice were purchased from Jackson Laboratory and were housed under controlled temperature (21 ± 1 °C) and humidity (35 ± 5%) conditions with a 12-h light/12-h dark period cycle, with ad libitum access to food and water. Mice were acclimated for one week before the start of the experiment. Eight- to ten-week-old female virgin mice were mated, and the presence of a vaginal plug was designated as gestational day 1 (GD1). Plugged dams were randomly assigned to one of 4 treatment arms and treated with AZT/3TC (100/50 mg/kg/day), ABC/3TC (100/50 mg/kg/day), TDF/FTC (50/33.3 mg/kg/day), or water as a control starting from GD1 through to GD15, which has previously been shown to yield levels similar to those seen in humans [36,37]. Drug tablets were purchased as prescription drugs and were pulverized and dissolved in sterile water. Drugs were administered to the dams daily by oral gavage based on body weight at the beginning of the experiment in a volume of ~100 µL/dam once. Animals were sacrificed at GD15, and heparinized blood was collected by cardiac puncture. Fetuses and placentas were collected and weighed prior to snap freezing in liquid nitrogen. Litter size and number of resorptions (residues from early fetal demise), as well as fetal and placental weight, were recorded. A total of 24 litters were collected for this study (6 control, 6 AZT/3TC, 5 ABC/3TC, and 7 TDF/FTC), with 2–4 randomly selected placentas from each litter being used for the expression analyses. A second cohort of 16 litters (4 per arm) was performed, with 2–4 randomly selected placentas from each litter being used for the mtDNA copy number analyses. 

### 2.2. Quantification of mtDNA Copy Number

Relative mtDNA copy number was determined using qPCR amplification of the mitochondrial genes ND1 and 16S rRNA and the nuclear gene hexokinase 2 (HK2), as previously described [38]. DNA was extracted from mouse placental tissue using an AllPrep DNA/RNA kit (Qiagen) according to the manufacturer’s instruction. Briefly, placental tissue was disrupted and homogenized in guanidine isothiocyanate-containing buffer to denature DNases and Rnases and facilitate the isolation of intact DNA. The homogenate was filtered, and DNA was selectively isolated using an AllPrep DNA spin column. The column was washed, and pure DNA was eluted. Samples were treated with Rnase A to digest remnant RNA. DNA concentration and purity were measured using a NanoDrop 2000c instrument, and only samples with a 260 to 280 nm ratio of ≥1.8 were selected for downstream real-time qPCR. Primer sequences for ND1, 16S, and HK2 are shown in Table 1. DNA was amplified in triplicate with SYBR Green I Master Mix (Roche, Cat#04887352001) and forward and reverse primers in a LightCycler 480 (Roche, Laval, QC, Canada). Analysis of the mtDNA/nDNA ratio was performed according to the ΔΔCt method, and the average of the ratios of ND1/HK2 and 16S/HK2 was used in the final analyses.

### 2.3. RNA Extraction and Real-Time qPCR

For gene expression analyses, total RNA was extracted from frozen (−70 °C) mouse placental tissues using a Life Technologies mirVana kit (Cat#AM-1560) according to the manufacturer’s instructions. Briefly, frozen placental tissue was processed immediately upon removal from the freezer without thawing. Placental tissue was weighed and homogenized in a denaturing lysis/binding buffer to prevent RNA degradation by Rnases using motorized tissue grinder. Homogenate was subjected to acid-phenol:chloroform extraction to produce semi-pure RNA. The final purification of total RNA was achieved by adding ethanol to the samples and passing the mixture through a filter cartridge containing a glass-fiber filter, which immobilized the RNA. The filter was washed a few times, and the RNA was eluted with a low-ionic-strength solution. RNA concentration and purity were measured using a NanoDrop 2000c instrument, and only samples with a 260 to 280 nm ratio of ≥2.0 were selected for downstream real-time qPCR. RNA samples were treated with Dnase I (Thermo Scientific-Cat#EN-0525), and cDNA was synthesized using an iScript cDNA synthesis kit (Bio-Rad-Cat#170-8891). 

Primers for POLG, cytochrome c-oxidase subunit II (COX-II), cytochrome c-oxidase subunit IV (COX-IV), PGC-1α, manganese superoxide dismutase (MnSOD), and citrate synthase (CS) were designed using NCBI primer blast (www.ncbi.nlm.nih.gov/tools/primer-blast/) and synthesized at the Center for Applied Genomics (Toronto, ON, Canada). Primer sequences are presented in Table 1. cDNA was amplified in triplicate with SYBR Green I Master Mix and forward and reverse primers in a LightCycler 480. Hypoxanthine-guanine phosphoribosyltransferase (HPRT) was used as housekeeping gene. The relative quantification (ΔΔCt) method was used for gene-expression content calculations [39].

### 2.4. Lipid Peroxidation (TBARS Assay)

Placental levels of the lipid peroxidation product malondialdehyde (MDA) were measured as a marker of lipid peroxidation and an indicator of oxidative stress using commercially available thiobarbituric-acid-reactive substance (TBARS) assay (R&D Systems, Minneapolis, MN, USA) and normalized to protein concentration. Tissue lysis and extraction were performed according to the manufacturer’s instructions. The assay spectrophotometrically measures the colored product produced from the reaction of MDA with thiobarbituric acid in the presence of acid and heat at 530–540 nm. Total protein concentration of the placenta lysate was quantified using the bicinchoninic acid (BCA) protein assay [40] with bovine serum albumin as standards. The data are presented as pmol of MDA per mg of protein. 

### 2.5. Statistical Analysis

Data were analyzed using Prism (GraphPad v8) and STATA v13. Normality of the data was assessed using the D’Agostino–Pearson omnibus normality test. Mixed-effects modelling was used to compare between treatments while accounting for litter effects, with treatment as a fixed effect and litter as a random variable. Correlation was assessed using Pearson’s correlation coefficient (r), or Spearman rank correlation coefficient (ρ). χ2 tests were used for categorical variables. Differences between treatments were considered statistically significant with a two-sided *p*-value < 0.05.

## 3. Results

### 3.1. Pregnancy Outcomes

Pregnant mice were administered AZT/3TC, ABC/3TC, or TDF/FTC suspended in water by gavage once daily starting at GD1 until sacrifice on GD15. Control mice were administered an equal volume of water. Litter size, number of resorptions, and fetal and placenta weights were recorded at GD15. Fetuses exposed to two of the three NRTIs were significantly smaller than controls (mean weight (SD), 0.230 g (0.048 g)), with the greatest growth restriction seen in the ABC/3TC group (0.163 g (0.035 g); *p* < 0.01), followed by the AZT/3TC group (0.176 g (0.036 g); *p* < 0.05; Figure 1a), whereas fetal weights in the TDF/FTC arm (0.198 g (0.044 g)) did not differ significantly from those of control mice. Fetuses exposed to TDF/FTC were significantly larger than fetuses exposed to ABC/3TC (*p* < 0.05; Figure 1a). Fetal weights were similar between the AZT/3TC and ABC/3TC groups.

Placental weights were also significantly lower in the AZT/3TC and ABC/3TC groups but not in the TDF/FTC group compared to the control group (mean weight (SD): control, 0.087 g (0.016 g); AZT/3TC, 0.069 g (0.019 g); ABC/3TC, 0.063 g (0.013 g); TDF/FTC, 0.076 g (0.018 g); Figure 1b). Placentas from the TDF/FTC group were significantly larger than placentas from the ABC/3TC group (*p* < 0.05). Placental weights did not differ significantly between AZT/3TC vs. ABC/3TC and AZT/3TC vs. TDF/FTC. 

Litter size did not differ significantly between the NRTI groups and the control group (mean (SD) control: 8.5 (0.5) fetuses; TDF/FTC: 7.0 (2.5) fetuses), although the AZT/3TC (6.4 (1.8) fetuses) and ABC/3TC (6.4 (1.9) fetuses) groups trended towards smaller litters. The fetal resorption rate was 10.9% for TDF/FTC, 17.9% for ABC/3TC, and was highest for AZT/3TC at a rate of 25.6% compared to 3.8% for the control group (*p* = 0.027 by χ2). Fetal viability was 100% in the control, TDF/FTC, and ABC/3TC groups and 97% in the AZT/3TC group (Figure 1c).

### 3.2. Placental mtDNA Content and Expression of POLG, COX-II, and Citrate Synthase Are Highest in Pregnant Mice on TDF/FTC

NRTI use has been shown to affect mtDNA and has been associated with mitochondrial toxicity, in part due to alterations in POLG [41]. We first assessed mtDNA relative content in the placenta by calculating the ratio of two mtDNA-encoded genes, ND1 and 16S, relative to the nuclear DNA-encoded gene HK2. The mtDNA/nDNA ratio was higher in the TDF/FTC group compared to the AZT/3TC group (*p* < 0.05) but not the ABC/3TC group. The mtDNA/nDNA ratio was similar between the AZT/3TC and ABC/3TC groups (Figure 2a). TDF/FTC was the only NRTI group to significantly differ from the control group, showing a higher mtDNA/nDNA ratio (*p* < 0.05).

We next assessed the effects of the three dual-NRTI backbones on placental expression of POLG. It is well documented that different NRTIs have varying inhibitory effects on POLG due to differences in their chemical structures [42]. We observed similar mRNA expression levels of POLG between the AZT/3TC, ABC/3TC, and control groups. In agreement with the mtDNA relative content findings, pregnant mice treated with TDF/FTC showed the highest placental expression of POLG (Figure 2b), which was significantly higher than that in the AZT/3TC (*p* < 0.05), ABC/3TC (*p* < 0.01), and control (*p* < 0.05) groups.

To further investigate the effect of dual-NRTIs on mitochondrial function and to assess the effect of alterations in mtDNA relative content and POLG expression, we measured the mRNA expression levels of the mtDNA-encoded gene cytochrome c-oxidase subunit II (COX-II) and the nuclear DNA (nDNA)-encoded gene COX-IV and computed the COX-II-to-COX-IV (COX-II:IV) ratio, which is an acceptable index of mitochondrial function (a decrease would suggest a downregulation of the enzymes necessary for mitochondrial function) [23]. The placental expression levels of COX-II were higher in the TDF/FTC group compared to the control (*p* < 0.001), AZT/3TC (*p* < 0.001), and ABC/3TC (*p* < 0.01; Figure 3a) groups, and in ABC/3TC compared to the control group (*p* < 0.05). However, no differences were observed between ABC/3TC and AZT/3TC, and AZT/3TC was similar to the control group. COX-IV levels were similar among all groups (Figure 3b). As with the COX-II data, the COX-II:IV ratio was higher in the TDF/FTC group compared to the AZT/3TC (*p* < 0.001; Figure 3C) group and compared to the ABC/3TC group (*p* < 0.05). TDF/FTC was the only treatment group that was significantly different from the control, with a higher COX-II:IV ratio (*p* < 0.01).

Declines in the levels of citrate synthase are also considered a marker of mitochondrial damage. Citrate synthase is often referred to as the pacemaker enzyme of the tricarboxylic acid cycle, and it is encoded by nuclear DNA. The level of citrate synthase is used as a marker of intact mitochondria [43,44,45]. Placental expression of citrate synthase in the TDF/FTC group was significantly higher compared to the AZT/3TC and ABC/3TC groups (*p* < 0.01; Figure 3d). No differences were observed between the AZT/3TC and ABC/3TC groups (Figure 3d). TDF/FTC was the only NRTI group that was significantly different from the control, showing higher levels of citrate synthase expression (*p* < 0.01).

### 3.3. Placental Levels of MDA Were Highest in AZT/3TC-Exposed Pregnant Mice

Mitochondrial toxicity and damage could result in generation of reactive oxygen species (ROS), driving lipid peroxidation and oxidative stress [46]. It has previously been shown that placentas from women exposed to AZT-containing regimens had elevated levels of ROS (measured as MDA levels) compared to controls [47]. To assess oxidative stress in the placenta, we measured placental levels of MDA, a marker of lipid peroxidation. In agreement with previous observations, pregnant mice treated with AZT/3TC had the highest levels of placental MDA (mean (SD): 3.19 (2.08) pmol/mg of protein) compared with ABC/3TC (0.62 (0.58) pmol/mg of protein; *p* < 0.001) and TDF/FTC (0.72 (0.29) pmol/mg of protein; *p* < 0.001; Figure 4). Placental MDA levels were similar between ABC/3TC- and TDF/FTC-treated groups and did not differ from MDA levels in the control group. Compared to the control group, only the AZT/3TC group showed a significant increase in MDA levels (*p* < 0.001), suggesting higher oxidative stress in this group.

### 3.4. Placental Expression of PGC-1α and MnSOD Are Highest in Pregnant Mice on TDF/FTC

PGC-1α is a nuclear-encoded mitochondrial protein that regulates mitochondria biogenesis, a process that can be stimulated to compensate for mitochondria damage [48]. We measured placental expression of PGC-1α as a marker of mitochondrial biogenesis. There was a stepwise increase in PGC-1α among the NRTI groups, with AZT/3TC exhibiting the lowest increase, followed by ABC/3TC and FTC/TDF with the highest increase (Figure 5a), resembling our findings of mtDNA content (see Figure 2a). Compared to the control group, TDF/FTC- (*p* < 0.001) and ABC/3TC (*p* < 0.05)-treated mice had significantly higher levels of PGC-1α expression. Mice in the TDF/FTC group also had significantly higher levels of PGC-1α compared to those in the AZT/3TC and ABC/3TC groups (*p* < 0.001).

PGC-1α is also a regulator of antioxidant responses, and a downregulation of the expression of PGC-1α has been shown to affect the expression of key antioxidant enzymes [49,50,51]. We investigated the downstream targets of PGC-1α by measuring the mRNA levels of MnSOD, the mitochondria-specific superoxide dismutase. Similar to the PGC-1α data, mice treated with TDF/FTC had significantly higher placental expression of MnSOD compared to mice treated with AZT/3TC (*p* < 0.01) and ABC/3TC (*p* < 0.05; Figure 5b); however, no differences were observed between AZT/3TC and ABC/3TC. Compared to the control group, only the TDF/FTC group was significantly different, with higher levels of MnSOD (*p* < 0.01). We also observed a significant correlation between PGC-1α and MnSOD levels in the NRTI groups (r = 0.49 (95%CI 0.20–0.70), *p* = 0.0021).

To investigate whether the levels of PGC-1α and MnSOD corresponded with reduced oxidative stress, we explored correlations between these factors and MDA levels. We observed a negative correlation between MnSOD and MDA levels (ρ = −0.59 (95%CI −0.83–−0.17), *p* = 0.008) and PGC-1α and MDA (ρ = −0.64 (95%CI −0.86–−0.23), *p* = 0.004) in the NRTI-treated groups, suggesting that upregulation of MnSOD and PGC-1α may serve to minimize oxidative stress.

### 3.5. Correlations with Fetal Weight and Resorption Rate

The placenta is a main contributor to fetal growth [52], and mitochondrial toxicity has been shown to contribute to placental insufficiency [53,54,55].

We examined the correlation between mitochondrial markers and fetal weight. In the mice exposed to dual NRTIs, placental COX-II expression, the COX-II:IV ratio, and PGC-1α were all directly correlated with fetal weight (r = 0.44 (95%CI 0.13–0.67), *p* = 0.0078 for COX-II; r = 0.44 (95%CI 0.13–0.67), *p* = 0.0073 for COX-II:IV ratio, and r = 0.34 (95%CI 0.02–0.60), *p* = 0.04 for PGC-1α). suggesting a protective effect on fetal growth with higher levels of COX-II, COX-II:IV ratio, and PGC-1α. 

We also investigated correlations between the assessed factors and resorption rate (early fetal loss). AZT/3TC had the highest resorption rate and the highest MDA levels, a marker of oxidative stress. We observed a positive correlation between MDA levels and resorption rate (ρ = 0.53 (95%CI 0.092–0.80), *p* = 0.018) and a negative correlation between PGC-1α (ρ = −0.44 (95%CI −0.67–−0.13), *p* = 0.006), as well as MnSOD (ρ = −0.50 (95%CI −0.71–−0.20), *p* = 0.0015). and resorption rate in the NRTI-exposed mice. These data may suggest that the oxidative stress observed in the AZT/3TC group may lead to higher resorption rates.

## 4. Discussion

Our data show that the three tested dual-NRTI backbones had different effects on fetal outcomes and markers of mitochondrial toxicity. Of the three regimens, TDF/FTC was the only dual-NRTI not associated with significant fetal growth restriction and had the lowest resorption rates. Placentas from mice treated with TDF/FTC had higher mtDNA relative content and higher mRNA levels of POLG, COX-II, citrate synthase, PGC-1α, and MnSOD compared to placentas of mice treated with AZT/3TC and ABC/3TC and compared to the control group. Compared to the control group, mice treated with AZT/3TC and ABC/3TC were associated with increased fetal growth restriction, although their mtDNA content and expression levels of POLG, COX-II, and citrate synthase were similar to those of the control group. PGC-1α levels were similar to those of the control group for AZT/3TC-treated mice but were higher for the ABC/3TC group. AZT/3TC was associated with high levels of lipid peroxidation, an indicator of oxidative stress, as well as the highest resorption rate.

A number of adverse outcomes have been associated with the use of ART containing NRTIs, including small for gestational age (SGA) births and low birth weight [56,57]. Our data suggest that TDF/FTC, AZT/3TC, and ABC/3TC have variable effects on fetal growth restriction in our model, with fetuses exposed to AZT/3TC or ABC/3TC being significantly smaller than fetuses in the control group and fetuses exposed to TDF/FTC being similar than those in the control group. During pregnancy, there is a huge metabolic burden on the placenta, which is supported by an increase in mitochondrial biogenesis and activities [58]; thus, mitochondrial dysfunction during this critical period could affect fetal and placental development [59]. In the NRTI-treated mice, we observed a positive correlation between fetal weight and COX-II levels, the COX-II:IV ratio (an index of mitochondria function) [23], and PGC-1α (a marker of mitochondria biogenesis). This may suggest that the upregulation of COX-II and PGC-1α seen in the TDF/FTC group is protective against fetal growth restriction, perhaps by compensating for NRTI mitotoxic effects by increasing mitochondrial numbers (also supported by the higher mtDNA relative content observed in the TDF/FTC group).

NRTI-induced mitochondrial toxicity has been shown to cause an increase in superoxide ions and induce oxidative stress [60,61,62,63,64,65,66]. Furthermore, altered mitochondrial function leads to increased production of free radicals and a compromised antioxidant defense system, resulting in oxidative damage to DNA, lipid peroxidation, and, consequently, oxidative stress [19]. It has previously been shown that placentas from women exposed to AZT-containing regimens had elevated levels of ROS (measured as MDA levels) compared to controls [47], although other studies have reported similar levels [23]. We also observed elevated levels of MDA in placentas from AZT/3TC-treated mice but not from placentas exposed to ABC/3TC or TDF/FTC. MDA levels correlated with number of resorptions, which were highest in the AZT/3TC group, but did not correlate with fetal weight. Oxidative stress is known to contribute to spontaneous abortion and recurrent pregnancy loss [54], suggesting that in the context of NRTI treatment, oxidative stress may be associated with early pregnancy loss, although not fetal growth restriction. 

MDA levels were negatively correlated with both MnSOD and PGC-1α levels in the NRTI-treated groups, suggesting that upregulation of MnSOD and/or PGC-1α may serve to minimize oxidative stress. Previous studies in non-pregnant mice treated with either AZT, 3TC, or TDF as single therapy reported higher plasma levels of 8-isoprostane, a marker of oxidative stress, with all three NRTIs, which was ameliorated with overexpression of MnSOD [67]. In our study, MnSOD was elevated in the non-AZT-containing regimens but was low in the AZT/3TC group. Why MnSOD levels failed to increase in the context of AZT exposure needs further investigation. One possibility may be that greater mitochondrial dysfunction in the AZT-exposed placentas prevented MnSOD production. 

This is the first study that compared the effects of dual-NRTI backbones of commonly prescribed ART on placental mitochondrial toxicity using a mouse pregnancy model. The limitations of our study include the lack of activity data for citrate synthase and MnSOD, a direct measure of mitochondrial function, and protein expression for the measured genes. Furthermore, we did not consider fetal sex in our analyses. Our study also does not include comparisons with the newer NRTI, tenofovir alafenamide.

## 5. Conclusions

In conclusion, our data suggest that compared to other dual-NRTI regimens, TDF/FTC was associated with the best fetal outcomes, namely higher fetal weight and the lowest resorption rates. Compared to the AZT/3TC and ABC/3TC groups, mice in the TDF/FTC group had higher mtDNA content and higher expression levels of placental POLG and COX-II, as well as higher placental levels of PGC-1α and MnSOD, which may help prevent mitochondrial toxicity by stimulating mitochondrial biogenesis and alleviating oxidative stress. Our findings also demonstrate that AZT-containing dual NRTIs were associated with higher oxidative stress, which correlated with higher resorption rates. Our data, although limited by being generated in an animal model, support the use of TDF/FTC in pregnancy over AZT/3TC.

## Figures and Tables

**Figure 1 pharmaceutics-14-01063-f001:**
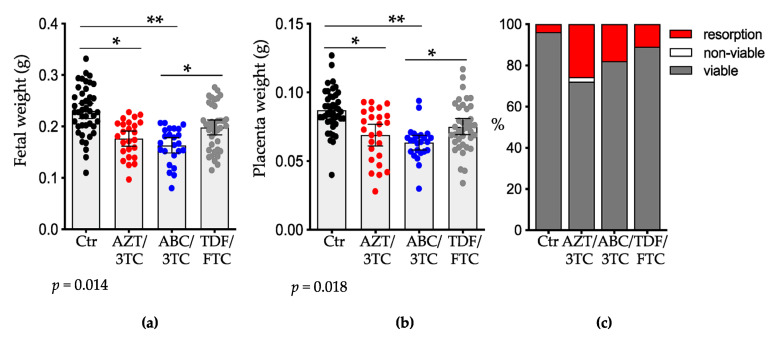
Maternal exposure to dual NRTIs is associated with lower fetal and placental weights. (**a**) Fetal weights; (**b**) placental weights; (**c**) viability and resorptions at gestational day 15 of mice treated with either AZT/3TC (red dots), ABC/3TC (blue dots), TDF/FTC (grey dots), or water (control; black dots). Data shown as scatter dot plots with mean and 95% confidence interval for (**a**,**b**). n = 43 fetuses/placentas from 6 litters for control; n = 25 fetuses/placentas from 6 litters for AZT/3TC; n = 24 fetuses/placentas from 5 litters for ABC/3TC; n = 40 fetuses/placentas from 7 litters for TDF/FTC. Statistical comparisons by mixed-effects modelling to control for litter effects (using treatment as a fixed effect and litter as a random effect). * = *p* < 0.05, ** = *p* < 0.01. AZT, zidovudine; 3TC, lamivudine; ABC, abacavir; FTC, emtricitabine; TDF, tenofovir.

**Figure 2 pharmaceutics-14-01063-f002:**
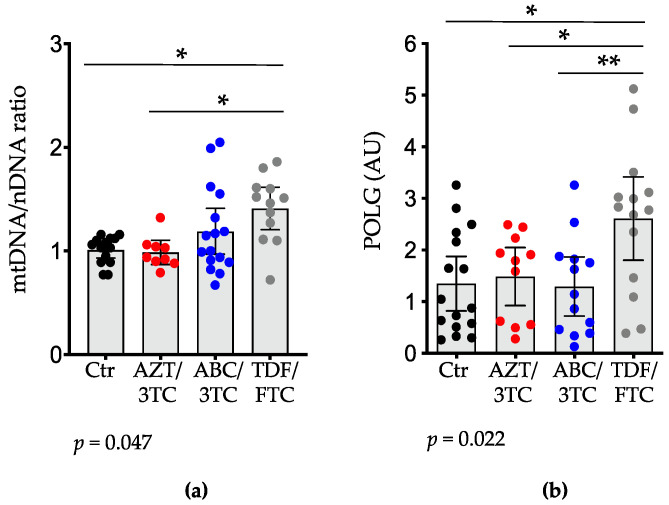
Placental mtDNA/nDNA ratio and mRNA levels of DNA polymerase gamma are elevated in pregnant mice treated with TDF/FTC. Mice were treated with either AZT/3TC (red dots), ABC/3TC (blue dots), TDF/FTC (grey dots), or water (control; black dots) throughout pregnancy. Placentas were collected at gestational day 15. (**a**) Placental mtDNA/nDNA, using ND1 and 16S RNA as the mitochondrial genes and HK2 as the nuclear gene. The average of the ratios of ND1/HK2 and 16S/HK2 is shown. n = 14 fetuses from 4 litters for control; n = 9 placentas from 4 litters for AZT/3TC; n = 16 placentas from 4 litters for ABC/3TC; n = 12 from 4 litters for TDF/FTC. (**b**) Placental mRNA expression levels of DNA polymerase gamma (POLG), with hypoxanthine-guanine phosphoribosyltransferase (HPRT) as the housekeeping gene. N = 16 placentas from 6 litters for control; n = 11 placentas from 6 litters for AZT/3TC; n = 13 placentas from 5 litters for ABC/3TC; n = 14 placentas from 6 litters for TDF/FTC. For both (**a**,**b**), data are scatter dot plots with mean and 95% confidence interval. Statistical comparisons by mixed-effects modelling to control for litter effects (using treatment as a fixed effect and litter as a random effect). * = *p* < 0.05, ** = *p* < 0.01. The *p*-values below the panels are for the model. AZT, zidovudine; 3TC, lamivudine; ABC, abacavir; FTC, emtricitabine; TDF, tenofovir; POLG, DNA polymerase gamma; AU, arbitrary units.

**Figure 3 pharmaceutics-14-01063-f003:**
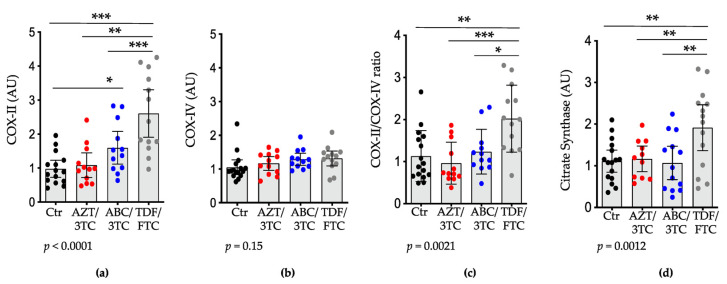
Placental mRNA levels of COX-II and citrate synthase are elevated in pregnant mice treated with TDF/FTC. Mice were treated with either AZT/3TC (red dots), ABC/3TC (blue dots), TDF/FTC (grey dots), or water (control; black dots) throughout pregnancy. Placentas were collected at gestational day 15. Placental mRNA expression of (**a**) COX-II, (**b**) COX-IV, (**c**) COX-II to COX-IV ratio; and (**d**) citrate synthase. Data are scatter dot plots with mean and 95% confidence interval (n = 16 placentas from 6 litters for control; n = 11–12 placentas from 6 litters for AZT/3TC; n = 12–13 placentas from 5 litters for ABC/3TC; n = 13–14 placentas from 6 litters for TDF/FTC). Statistical comparisons by mixed-effects modelling to control for litter effects (using treatment as a fixed effect and litter as a random effect). * = *p* < 0.05, ** = *p* < 0.01, *** = *p* < 0.001. The *p*-values below the panels are for the model. AZT, zidovudine; 3TC, lamivudine; ABC, abacavir; FTC, emtricitabine; TDF, tenofovir; COX-II, cytochrome c-oxidase subunit-II; COX-IV, cytochrome c-oxidase subunit IV; AU, arbitrary units.

**Figure 4 pharmaceutics-14-01063-f004:**
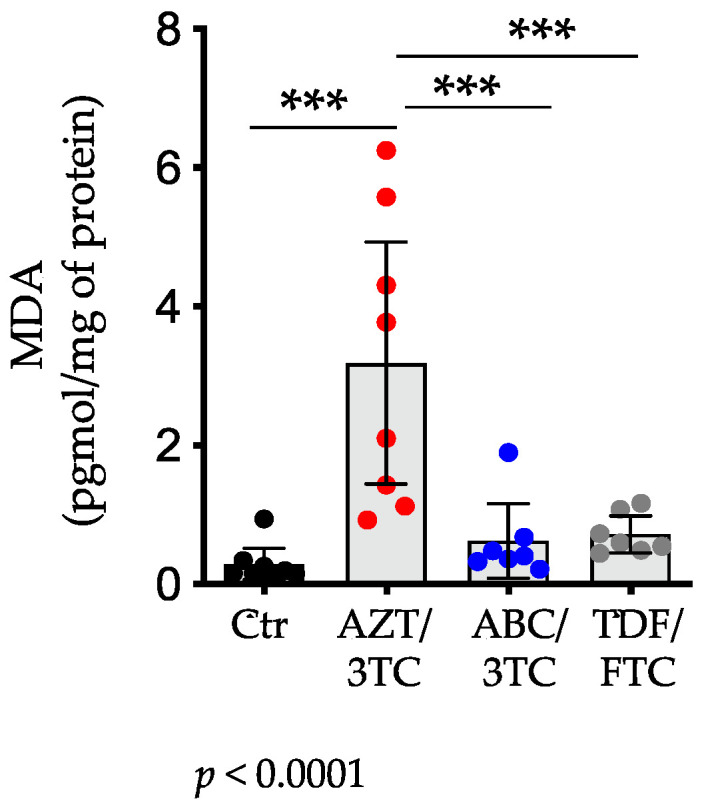
Placental MDA levels are elevated in pregnant mice treated with AZT/3TC. Placental concentrations of MDA of mice exposed to AZT/3TC (red dots), ABC/3TC (blue dots), TDF/FTC (grey dots), or water (control; black dots). Data are scatter dot plots with mean and 95% confidence interval (n = eight placentas from four litters for control; n = four placentas from six litters for AZT/3TC; n = seven placentas from four litters for ABC/3TC; n = seven placentas from four litters for TDF/FTC). Statistical comparisons by mixed-effects modelling to control for litter effects (using treatment as a fixed effect and litter as a random effect). *** = *p* < 0.001. The *p*-value below the panel is for the model. AZT, zidovudine; 3TC, lamivudine; ABC, abacavir; FTC, emtricitabine; TDF, tenofovir; MDA, malondialdehyde.

**Figure 5 pharmaceutics-14-01063-f005:**
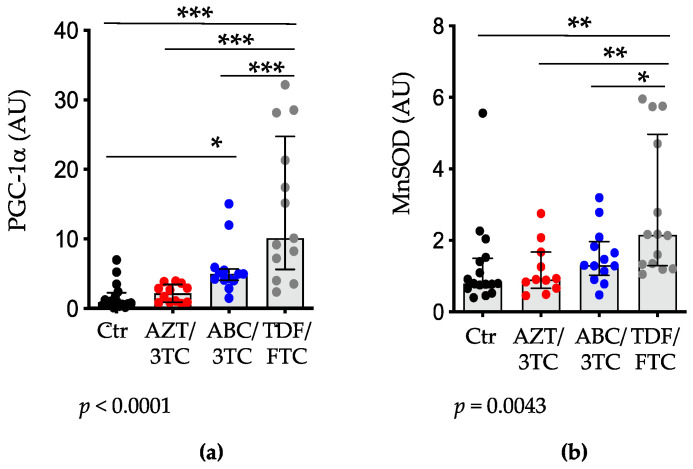
Placental mRNA levels of PGC-1α and MnSOD are elevated in pregnant mice treated with TDF/FTC. Placental mRNA expression levels of (**a**) PGC-1α and (**b**) MnSOD of mice treated with AZT/3TC (red dots), ABC/3TC (blue dots), TDF/FTC (grey dots), or water (control; black dots). Data are scatter dot plots with median and interquartile range (n = 16 placentas from 6 litters for control; n = 11–12 placentas from 6 litters for AZT/3TC; n = 12–13 placentas from 5 litters for ABC/3TC; n = 13–14 placentas from 6 litters for TDF/FTC). Statistical comparisons by mixed-effects modelling to control for litter effects (using treatment as a fixed effect and litter as a random effect). * = *p* < 0.05, ** = *p* < 0.01, *** = *p* < 0.001. The *p*-values below the panels are for the model. AZT, zidovudine; 3TC, lamivudine; ABC, abacavir; FTC, emtricitabine; TDF, tenofovir; PGC-1α, peroxisome proliferator-activated receptor gamma coactivator 1-alpha; MnSOD, manganese superoxide dismutase.

**Table 1 pharmaceutics-14-01063-t001:** Sequence of primers used for qPCR analyses.

Gene ^1^	Primer (Sense)	Primer (Anti-Sense)
ND1	CTAGCAGAAACAAACCGGGC	CCGGCTGCGTATTCTACGTT
16S rRNA	CCGCAAGGGAAAGATGAAAGAC	TCGTTTGGTTTCGGGGTTTC
HK2	GCCAGCCTCTCCTGATTTTAGTGT	GGGAACACAAAAGACCTCTTCTGG
POLG	GCAGGATGGGCAGGAACA	GCATCCGGGAGTCCTGAA
CS	CAGCAGTATCGGAGCCATTGA	GGGTCGGTGTAGCCTAACAT
PGC-1α	GCCGTGTGATTTACGTTGGTAA	AAAACTTCAAAGCGGTCTCTCAA
MnSOD	CTGGAGCCACACATTAACGC	CGGTGGCGTTGAGATTGTTC
COX-II	AACCGAGTCGTTCTGCCAAT	CTAGGGAGGGGACTGCTCAT
COX-IV	TTCACTGCGCTCGTTCTGAT	CACCCAGTCACGATCGAAAGTA
HPRT	AGCGTCGTGATTAGCGATGA	ACACTTTTTCCAAATCCTCGGC

^1^ ND1, mitochondrially encoded NADH:ubiquinone oxidoreductase core subunit 1; HK2, hexokinase 2; POLG, DNA polymerase gamma; CS, citrate synthase; PGC-1α, peroxisome proliferator-activated receptor gamma coactivator 1-alpha; MnSOD, manganese superoxide dismutase; COX-II, cytochrome c-oxidase subunit II; COX-IV, cytochrome c-oxidase subunit IV; HRPT, hypoxanthine-guanine phosphoribosyltransferase.

## Data Availability

All relevant data are presented in the paper. For any additional information, please contact the corresponding author.

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
