# Peer review of "Comparison of the Effects of Three Dual-Nucleos(t)ide Reverse Transcriptase Inhibitor Backbones on Placenta Mitochondria Toxicity and Oxidative Stress Using a Mouse Pregnancy Model"

_pharmaceutics, 2022, doi:10.3390/pharmaceutics14051063_

Round 1

Reviewer 1 Report

Comments:

  The authors have compared the influence on the placental mitochondrial toxicity through three types of dual-nucleoside reverse transcriptase inhibitors (NRTIs) backbones in order to determine the least toxic combination associated with pregnancy. In general, the whole experiment seems very easy and clear. However, the biggest problem is that your experiments are far from innovative and crude. In my opinion, it should be more detailed, comprehensive and attentive on experimental scheme design. Therefore, I’m sorry to tell you this manuscript couldn’t be approved.

Q1: The experimental data are too simple to support your conclusion. 

Q2: The language description of Materials and Methods section is too vague and does not meet the requirements of experimental details are provided to allow your work will be reproduced by anyone.

Q3: The innovation and comprehensiveness of your experimental design is still needed much improvements.

Author Response

The authors have compared the influence on the placental mitochondrial toxicity through three types of dual-nucleoside reverse transcriptase inhibitors (NRTIs) backbones in order to determine the least toxic combination associated with pregnancy. In general, the whole experiment seems very easy and clear. However, the biggest problem is that your experiments are far from innovative and crude. In my opinion, it should be more detailed, comprehensive and attentive on experimental scheme design. Therefore, I’m sorry to tell you this manuscript couldn’t be approved.

Q1: The experimental data are too simple to support your conclusion. 

Thank you for your comments.  While we agree that none of our experiments are very complicated, they were performed well and reproducible, and the conclusions drawn from the findings are conservative. We stated the limitations of the study and have revised the manuscript as per the reviewers’ recommendations. 

Q2: The language description of Materials and Methods section is too vague and does not meet the requirements of experimental details are provided to allow your work will be reproduced by anyone.

Thank you for highlight this. We have added additional details to our methods (see revised manuscript – all sub-sections of the methods have been revised to include more details).

Q3: The innovation and comprehensiveness of your experimental design is still needed much improvements.

The scope of our study is not big, but our objectives are clearly stated, our conclusions are not overstated, and our limitations are clearly identified.  The innovation of our study is that we present novel data not previously reported addressing an important question - a direct comparison of 3 dual-NRTI regimens in the context of pregnancy. Our study merits publication because of the importance of the question asked, and adds to the published literature. 

As fellow scientists we would also like to highlight the merit of “less innovative” studies, such as replication studies and studies that provide incremental steps in our understanding of processes, which are the mainstay of scientific progress.   

Reviewer 2 Report

This is an interesting manuscript addressing a relevant issue.

However I have major concerns regrading the experimental design and data treatment.

It is apparent from the various figures in the manuscript that the authors considered each placenta as an experimental point. I am afraid that this is not correct and an exercise of pseudo-replication. Placentas from the same dam are not by any means independent and cannot be treated as such. The litter should be the experimental unit. So if you have 5 dams per experimental group (the number is not mentioned in the methods but it should be), you should have 5 points per experimental group not up to 43 as mentioned in Figure 1...

So all the analysis needs to be re-done in the light of this assumption: the litter is the experimental unit. I'll be glad to read this manuscript once this is corrected.

Other minor comments

What was the number of dams per experimental group?

Why was sampling done at GD15 and not around GD20 when pregnancy is close to termination?

Author Response

This is an interesting manuscript addressing a relevant issue. However I have major concerns regrading the experimental design and data treatment. It is apparent from the various figures in the manuscript that the authors considered each placenta as an experimental point. I am afraid that this is not correct and an exercise of pseudo-replication. Placentas from the same dam are not by any means independent and cannot be treated as such. The litter should be the experimental unit. So if you have 5 dams per experimental group (the number is not mentioned in the methods but it should be), you should have 5 points per experimental group not up to 43 as mentioned in Figure 1...

So all the analysis needs to be re-done in the light of this assumption: the litter is the experimental unit. I'll be glad to read this manuscript once this is corrected.

This is an excellent point.  However, since we present correlation analyses between the placental markers measured and fetal weight, we felt it would be helpful and necessary to present all the fetal and placenta weights. To account for intra-litter similarity we have revised our analyses to utilize a mixed effects analysis method, which includes litter as a random variable, to account for the intra-litter similarity expected.  This has altered our findings a bit for figure 1 – now TDF/FTC fetal and placental weight is no longer significantly different from the control group. 

For all other figures (other than figure 1), we use only a portion of the placentas from each litter – 2-4/litter.  These were randomly selected.  Again, we have redone our statistical comparison using mixed effects models to account for litter similarity.  Our findings remain as before.

We have also clarified the number of litters and number of placentas included in the figure legends.

Other minor comments

What was the number of dams per experimental group?

We have added this information to the methods section (line 108-112 of revised manuscript), and have expanded the details of the number of dams and number of placentas included for each figure in the figure legends.  We hope this improves the clarity.

Why was sampling done at GD15 and not around GD20 when pregnancy is close to termination?

At GD15 the placenta is fully formed in C57Bl/6 mice and this time point is at the beginning of a huge growth burst in the developing fetus.  Thus, changes identified in the placenta at this time point are likely to predict poor fetal growth later in gestation – so this is an informative time point for identifying NRTI-associated dysregulation in the placenta that could predict poor fetal growth.  

Reviewer 3 Report

  1. It seems to be better that 'p<0.01' be changed to 'p<0.01 (italic style)' in the manuscript. The same applies to other pages.
  2. In line 95, '100ul/dam' be changed to '100 ul/dam''.
  3. Some of the experimental results shown in this manuscript seem to conflict with each other. For example, in Figure 1, TDF/FTC group shows a significant decrease in fetal weight and placenta weight compared to the control group, and in Figure 3, TDF/FTC group shows a significant increase in Citrate Synthase mRNA level compared to the control group. However, as shown in Figure 4, there was no significant change in the level of MDA compared to the control group. A more detailed review of these areas is needed.

Author Response

  1. It seems to be better that 'p<0.01' be changed to 'p<0.01 (italic style)' in the manuscript. The same applies to other pages.

We have made the requested changes throughout the manuscript.

  1. In line 95, '100ul/dam' be changed to '100 ul/dam''.

We have made the requested changes throughout the manuscript.

  1. Some of the experimental results shown in this manuscript seem to conflict with each other. For example, in Figure 1, TDF/FTC group shows a significant decrease in fetal weight and placenta weight compared to the control group, and in Figure 3, TDF/FTC group shows a significant increase in Citrate Synthase mRNA level compared to the control group. However, as shown in Figure 4, there was no significant change in the level of MDA compared to the control group. A more detailed review of these areas is needed.

Following reviewers’ recommendation, we have re-analysed our data using a mixed effects method to account for litter. After making this adjustment we no longer see a significant difference in fetal weight or placenta weight between TDF/FTC and control.  In all cases TDF/FTC appears to be the dual-NRTI associated with the best fetal outcomes.  The upregulation we see across the board in markers of mitochondria number and function (including higher mtDNA/nDNA ratio, and higher levels of POLg, COX-II, and CS) may suggest that this may be part of a compensatory mechanism induced in the TDF/FTC treated mice that may contribute to the maintenance of mitochondria function in the placenta and thus fetal growth. This is in agreement with our observation of the least fetal growth restriction in the TDF/FTC group. As for associations with MDA levels – lipid peroxidation can result from a variety of processes including mitochondria toxicity resulting in generation of excess ROS. The higher levels of MnSOD and PGC-1a observed in the TDF/FTC group may have provided some protection from lipid peroxidation resulting in MDA levels being similar to control. As well, mitochondrial damage/toxicity has multifactorial causes and not just oxidative stress. In our study, CS was used as a marker of mitochondrial damage and intact mitochondria. The increased expression of CS in the TDF/FTC may be associated with mitochondrial proliferation which was not observed in the control. A limitation of our study, mentioned in our manuscript, is that we did not measure the activity of CS which could show a totally different profile.

Reviewer 4 Report

In this article Balogun and Serghides aim to assess nucleoside / nucleotide reverse transcriptase inhibitors (NRTIs) that are the commonly used dual backbones of HIV antiretroviral therapy (ART). The authors state that ART use in pregnancy has been associated with adverse birth outcomes in part due to NRTI-induced mitochondrial toxicity and that there is a lack of comparable data when looking at the effects of the commonly prescribed dual-NRTI regimens on placental mitochondria toxicity in pregnancy. In the current article, Balogun and Serghides compared the dual combinations of zidovudine / lamivudine, abacavir / lamivudine, and tenofovir / emtricitabine in a murine model, and examined the markers of placental mitochondria function and oxidative stress.

NRTI- induced mitochondrial toxicity in the placenta during pregnancy is a major concern because NRTIs are permeable to the placenta. It is evident that efficient mitochondrial function is vital to pregnancy and fetal development, and mitochondrial toxicity during pregnancy has the ability to significantly alter the course of pregnancy and lead to serious adverse pregnancy outcomes. These can include fetal growth restriction which is an extremely unwanted outcome.

Under the experimental conditions used in this study, the authors suggest that NRTIs were associated with lower fetal and placenta weights compared to the relevant controls, with tenofovir / emtricitabine being associated with the least fetal and placenta weight reduction and lower resorption rates. Placental mitochondria DNA content and placental expression of cytochrome c-oxidase subunit-II, DNA polymerase gamma, and citrate synthase were higher in tenofovir / emtricitabine-treated mice compared to the other groups. Zidovudine / lamivudine-treated mice had elevated malondialdehyde levels (with this being an oxidative stress marker) compared to other groups. The authors also stated that the zidovudine / lamivudine-treated mice had lower mRNA levels of manganese superoxide dismutase and peroxisome proliferator-activated receptor gamma coactivator 1-alpha in the placenta compared to the tenofovir / emtricitabine-treated mice.

Balogun and Serghides state that they observed some differences between NRTI regimens on placental mitochondrial function and birth outcomes. The authors go on to conclude that their data suggest that tenofovir / emtricitabine was associated with larger fetuses, increased mitochondrial DNA content, and higher expression of mitochondrial-specific antioxidant enzymes as well as mitochondrial biogenesis enzymes, whereas zidovudine / lamivudine was apparently associated with markers of placental oxidative stress.

Main points and comments:

  1. The article is well-written, easy to read and presents some very nice, compelling data.
  2. How was it decided to dose at 100µl/dam?
  3. What is the bioavailability of the drugs when they are administered to mice in this way?
  4. The authors state they exposed the dams to “….human-relevant plasma concentrations of AZT/3TC, ABC/3TC, or FTC/TDF….”.  How were the relevant doses decided upon and were any other concentrations tested within the study? Do the authors have other preliminary data to support why they chose the doses they have used in this article?
  5. Throughout the manuscript the authors refer to tenofovir / emtricitabine which they subsequently call TDF / FTC from line 74. These 2 compounds are always referred to in the order shown ie tenofovir / emtricitabine (TDF / FTC) except for the opening section of the Materials and Methods (section 2.1) where the order has changed to FTC / TDF when the dosing regimen is discussed. Is this the correct order as shown on line 93 or is this back to front? Was the FTC administered at 33.3 mg/kg/day? Please can the authors clarify and make the order consistent throughout the paper?
  6. The authors state that mice were sacrificed at GD15. Were any other time points tested? Would the authors expect to generate even more significant differences at slightly later time points?
  7. Were the authors surprised that there was little effect on litter size in this study?
  8. Figure 2 demonstrates some very interesting data sets with some statistically significant differences between the different treatment groups and the controls.
  9. Are there other systems available to the authors to assess mitochondrial function other than the COX-II:IV ratio? Do they have data from other assay systems? How reliable / robust are the COX-II:IV ratios?
  10. I find Figure 3 slightly misleading as the axes are very different for (a), (b) and (c).  I understand why they have been presented in this way but it may be better and less misleading to show (b) on the same scale as (a). Of course, this is my personal preference and I fully appreciate there are other views. It is just a bit odd when you look at the ratios in (c) and find highly significant ratios when at first glance there is apparently very little difference in the TDF / FTC values in (a) and (b).
  11. Were the authors surprised by the data generated in Figure 4?
  12. The authors have demonstrated different effects on fetal outcomes and markers of mitochondrial toxicity by comparing three dual-NRTI backbones in this study. They have been able to confirm that the TDF / FTC regimen was associated with the least fetal growth restriction and lowest resorption rates.
  13. I fully appreciate that the authors do state themselves (line 375), that one limitation of the study is the fact that this work has been carried out in an animal model but I also appreciate that the work has to start somewhere. The data are quite convincing and there are statistically significant results that are worthy of being presented to the scientific community. The authors do explain the limitations (lines 361 to 365) and so they are aware of the shortcomings, but they do still have reasonable results to present.
  14. Just as a comment, the majority of the references used in this study are quite old. Can the authors comment on this please? Presumably this highlights how little recent work has been carried out in this field of research?
  15. Minor issue: There is the odd word missing (i.e. Line 349).

I do think the authors have made a good attempt to fulfil the aims of their study which were to compare the effects of some commonly prescribed dual-NRTI backbones in pregnancy by studying zidovudine / lamivudine (AZT/3TC), abacavir / lamivudine (ABC/3TC), and tenofovir / emtricitabine (TDF/FTC) on placental mitochondrial toxicity. This was neatly done by assessing the impact of these NRTI combinations on mitochondrial DNA content, and analysing the genes involved in the mitochondria-mediated antioxidant defence system, mitochondrial biogenesis, and mitochondrial function as markers of mitochondrial toxicity. The authors state that the objective of their study was to directly compare these NRTI backbones in a murine pregnancy model with the goal of determining which combination was associated with the best pregnancy outcomes and the least mitochondria toxicity.  I do appreciate that they have done just that and have generated some interesting results that should be able to be extrapolated back to the human situation even though the generated results are from a murine model. Obviously, there are going to be some differences, but I would think that these results help to lay a foundation for more detailed analysis in the human population as this is an important area of research to take forward.

Author Response

Under the experimental conditions used in this study, the authors suggest that NRTIs were associated with lower fetal and placenta weights compared to the relevant controls, with tenofovir / emtricitabine being associated with the least fetal and placenta weight reduction and lower resorption rates. Placental mitochondria DNA content and placental expression of cytochrome c-oxidase subunit-II, DNA polymerase gamma, and citrate synthase were higher in tenofovir / emtricitabine-treated mice compared to the other groups. Zidovudine / lamivudine-treated mice had elevated malondialdehyde levels (with this being an oxidative stress marker) compared to other groups. The authors also stated that the zidovudine / lamivudine-treated mice had lower mRNA levels of manganese superoxide dismutase and peroxisome proliferator-activated receptor gamma coactivator 1-alpha in the placenta compared to the tenofovir / emtricitabine-treated mice.

Balogun and Serghides state that they observed some differences between NRTI regimens on placental mitochondrial function and birth outcomes. The authors go on to conclude that their data suggest that tenofovir / emtricitabine was associated with larger fetuses, increased mitochondrial DNA content, and higher expression of mitochondrial-specific antioxidant enzymes as well as mitochondrial biogenesis enzymes, whereas zidovudine / lamivudine was apparently associated with markers of placental oxidative stress.

Main points and comments:

1. The article is well-written, easy to read and presents some very nice, compelling data.

Thank you for your kind assessment.

2. How was it decided to dose at 100µl/dam?

The actual dose of drug administered to each mouse was calculated based on body weight at the beginning of the experiment (~20g).  The appropriate dose is administered in ~100ul of water.  The drug is suspended in the water.  100ul is an ethically acceptable amount of liquid to administer by gavage to a mouse, and is sufficient to allow for accurate measurement and administration of the drug dose.

3. What is the bioavailability of the drugs when they are administered to mice in this way?

Based on available literature the bioavailability of TDF is about 20-40% (higher if taken with a high fat meal), AZT is about 50-75%, and 3TC, ABC, and FTC is about 80% in both mice and humans when administered orally.

4. The authors state they exposed the dams to “….human-relevant plasma concentrations of AZT/3TC, ABC/3TC, or FTC/TDF….”.  How were the relevant doses decided upon and were any other concentrations tested within the study? Do the authors have other preliminary data to support why they chose the doses they have used in this article?

We have published findings from our mouse model optimization studies looking at various drug concentration in pregnant mice and the resulting plasma Cmax and Cmin levels, as well as amniotic fluid levels (Kala S et al. 2018), and for ABC and 3TC also fetal brain concentrations of the drugs (Gilmore JC et al. 2021).  The Kala et al. reference was already included in the manuscript (ref #37).  We have now added the Gilmore et al. reference as well (ref #38 in revised manuscript).

Kala S, Watson B, Zhang JG, Papp E, Guzman Lenis M, Dennehy M, Cameron DW, Harrigan PR, Serghides L. Improving the clinical relevance of a mouse pregnancy model of antiretroviral toxicity; a pharmacokinetic dosing-optimization study of current HIV antiretroviral regimens. Antiviral Res. 2018 Nov;159:45-54. PMID: 30236532.

Gilmore JC, Zhang G, Cameron DW, Serghides L, Bendayan R. Impact of in-utero antiretroviral drug exposure on expression of membrane-associated transporters in mouse placenta and fetal brain. AIDS. 2021 Nov 15;35(14):2249-2258. PMID: 34175869.

5. Throughout the manuscript the authors refer to tenofovir / emtricitabine which they subsequently call TDF / FTC from line 74. These 2 compounds are always referred to in the order shown ie tenofovir / emtricitabine (TDF / FTC) except for the opening section of the Materials and Methods (section 2.1) where the order has changed to FTC / TDF when the dosing regimen is discussed. Is this the correct order as shown on line 93 or is this back to front? Was the FTC administered at 33.3 mg/kg/day? Please can the authors clarify and make the order consistent throughout the paper?

Thank you for catching this discrepancy.  We have corrected the mention in the methods to be TDF/FTC.  The dose stated is correct – FTC is administered at 33.3mg/kg/day and TDF is administered at 50mg/kg/day.

6. The authors state that mice were sacrificed at GD15. Were any other time points tested? Would the authors expect to generate even more significant differences atslightly later time points?

We chose GD15 as at this time point the placenta is fully formed in C57Bl/6 mice, and this time point is at the beginning of a huge growth burst in the developing fetus. Thus, changes identified in the placenta at this time point are likely to predict poor fetal growth later in gestation – so this is an informative time point for identifying NRTI-associated dysregulation in the placenta that could predict poor fetal growth.  In previous studies we have examined the impact of combination antiretroviral regimen on placental vasculature at both GD15 and GD18 and saw similar changes at both time points.  It is likely that we may have observed bigger differences at later time points, but any changes seen at this time point, which precedes the big fetal growth spurt, are likely to be more clinically important.  

7. Were the authors surprised that there was little effect on litter size in this study?

Litter size was smaller in ABC/3TC and AZT/3TC vs. control but this did not reach significance.  This may be a sample size issue, although we have consistently observed minimal effects on litter size in our mouse experiments using a wide variety of antiretrovirals.  We did observe higher resorption rates particularly in the AZT/3TC group.

8. Figure 2 demonstrates some very interesting data sets with some statistically significant differences between the different treatment groups and the controls.

Thank you.

9. Are there other systems available to the authors to assess mitochondrial function other than the COX-II:IV ratio? Do they have data from other assay systems? How reliable / robust are the COX-II:IV ratios?

We recognize that the COX-II:IV ratio is not the best metric to assess mitochondria function. The aim of our study was to assess mitochondria function using surrogate markers. The strength of our data is the consistency between the various factors assessed, i.e. TDF/FTC exhibits higher mtDNA/nDNA ratio, higher POLG levels, higher COX-II and COX-II:IV ratio, higher CS levels, and higher PGC-1a and MnSOD levels. Moving forward, we are planning to perform metabolic monitoring studies in mice using the Comprehensive Lab Animal Monitoring System, which will provide us with an in vivo functional metric of oxygen consumption. Cytometric assessments using probes are also available for assessing mitochondria function. 

10. I find Figure 3 slightly misleading as the axes are very different for (a), (b) and (c).  I understand why they have been presented in this way but it may be better and less misleading to show (b) on the same scale as (a). Of course, this is my personal preference and I fully appreciate there are other views. It is just a bit odd when you look at the ratios in (c) and find highly significant ratios when at first glance there is apparently very little difference in the TDF / FTC values in (a) and (b).

We have presented figure 3b on the same scale as figure 3a.  Thank you for the suggestion, we agree this improves clarity.

11. Were the authors surprised by the data generated in Figure 4?

The high MDA levels in the AZT/3TC group were not surprising – similar findings have been observed in pregnant women. However, the degree of difference was surprising.  We were also slightly surprised to see such minimal increases in MDA in the other 2 dual-NRTI regimens, although we were reassured by our MnSOD and PGC-1a data that corroborated the MDA findings.

12. The authors have demonstrated different effects on fetal outcomes and markers of mitochondrial toxicity by comparing three dual-NRTI backbones in this study. They have been able to confirm that the TDF / FTC regimen was associated with the least fetal growth restriction and lowest resorption rates.

Thank you.

13. I fully appreciate that the authors do state themselves (line 375), that one limitation of the study is the fact that this work has been carried out in an animal model but I also appreciate that the work has to start somewhere. The data are quite convincing and there are statistically significant results that are worthy of being presented to the scientific community. The authors do explain the limitations (lines 361 to 365) and so they are aware of the shortcomings, but they do still have reasonable results to present.

Thank you.

14. Just as a comment, the majority of the references used in this study are quite old. Can the authors comment on this please? Presumably this highlights how little recent work has been carried out in this field of research?

We have gone back and rechecked for newer references but were not able to find any additional relevant references. We agree with the reviewer that this highlights how little recent work has been carried out in this field.

15. Minor issue: There is the odd word missing (i.e. Line 349).

Thank you for picking that up.  We have added the missing “be” on that line.

I do think the authors have made a good attempt to fulfil the aims of their study which were to compare the effects of some commonly prescribed dual-NRTI backbones in pregnancy by studying zidovudine / lamivudine (AZT/3TC), abacavir / lamivudine (ABC/3TC), and tenofovir / emtricitabine (TDF/FTC) on placental mitochondrial toxicity. This was neatly done by assessing the impact of these NRTI combinations on mitochondrial DNA content, and analysing the genes involved in the mitochondria-mediated antioxidant defence system, mitochondrial biogenesis, and mitochondrial function as markers of mitochondrial toxicity. The authors state that the objective of their study was to directly compare these NRTI backbones in a murine pregnancy model with the goal of determining which combination was associated with the best pregnancy outcomes and the least mitochondria toxicity.  I do appreciate that they have done just that and have generated some interesting results that should be able to be extrapolated back to the human situation even though the generated results are from a murine model. Obviously, there are going to be some differences, but I would think that these results help to lay a foundation for more detailed analysis in the human population as this is an important area of research to take forward.

We thank you for your kind comments.

Round 2

Reviewer 1 Report

I am satisfied with the revisions that have been made and would be happy to recommend for publication.

Author Response

Thank you for your comments.  They have dramatically improved our manuscript.

Reviewer 2 Report

The authors satisfactorily addressed my comments.

Author Response

(The authors gave the same response as above.)
